# EucliFold: Probing 3D Euclidean Prior in VLMs via Cognitively-Stratified Folding Tasks

## Abstract

Humans leverage robust 3D spatial priors to align perception with the physical world, enabling flexible and intelligent behavior. While Vision-Language Models (VLMs) exhibit impressive zero-shot performance, it remains unclear whether they possess genuine spatial reasoning capabilities, as standard evaluations are confounded by dataset bias and spurious correlations. To address this, we introduce **EucliFold**, a synthetic visual question-answering benchmark focused on cube net folding in Euclidean space—a domain that enables precise analysis while requiring genuine spatial understanding. We propose a **cognitively-stratified evaluation framework** that decomposes spatial reasoning into three hierarchical levels: **Perception** (grounding sensory input to spatial representations), **Operation** (manipulating representations according to instructions), and **Imagination** (autonomous spatial problem-solving under geometric constraints). This decomposition isolates genuine spatial reasoning from superficial pattern matching. To mitigate evaluation biases, we employ **Winograd-style accuracy** using minimal-pair contrastive samples. Our evaluation reveals that state-of-the-art VLMs demonstrate reasonable perceptual capabilities but fail significantly at operational and imagination-level spatial reasoning, suggesting reliance on statistical patterns rather than genuine geometric understanding. Detailed analysis on the inconsistent performance distribution of models shows that some SOTA VLMs can use partial strategies to solve some cases but fail to adopt generalizable solution, confirming the effectiveness of our cognitively-stratified decomposition. EucliFold provides a rigorous testbed for probing emergent spatial priors in future models and demonstrates how systematic cognitive decomposition can reveal nuanced capability gaps in VLMs.

## 1 Introduction

Vision-Language Models (VLMs) have demonstrated remarkable zero-shot generalization capabilities across diverse tasks (Liu et al., 2023b; Chen et al., 2024), suggesting the emergence of sophisticated internal representations and reasoning mechanisms. Recent research has provided compelling evidence that transformer-based models (Vaswani et al., 2017) can develop internal representations that align with real-world structure (Gurnee & Tegmark, 2023) and human perception (Abdou et al., 2021; Huh et al., 2024), learn generalizable solutions (Zhong et al., 2023; Huang et al., 2024), and exhibit emergent behaviors (Brown et al., 2020; Wei et al., 2022). However, recent studies suggest that the spatial reasoning capabilities of VLMs remain relatively weak, compared with human (Ma et al., 2024; Liu et al., 2023a; Tang et al., 2025; Tong et al., 2025). Current spatial ability evaluation datasets struggle to quantitatively assess whether VLMs possess generalizable spatial priors. This study explores the quantitative measurement of such priors.

It is hard to determine if a VLM solve spatial reasoning problem through emergent spatial ability or fixed strategies it learnt from similar situation. Also, it is hard to claim a VLM do not have certain emergent spatial ability. To address these challenges, we propose a evaluation framework grounded in cognitive science principles. We introduce a theoretically motivated decomposition of spatial prior into three qualitatively distinct levels: **Perception** (grounding sensory input to spatial repre-

sentations), **Operation** (manipulating spatial representations according to external instructions), and **Imagination** (autonomous spatial problem-solving under geometric constraints). This hierarchical framework enables precise characterization of VLMs' spatial capabilities and identifies the specific cognitive levels at which generalization succeeds or fails.

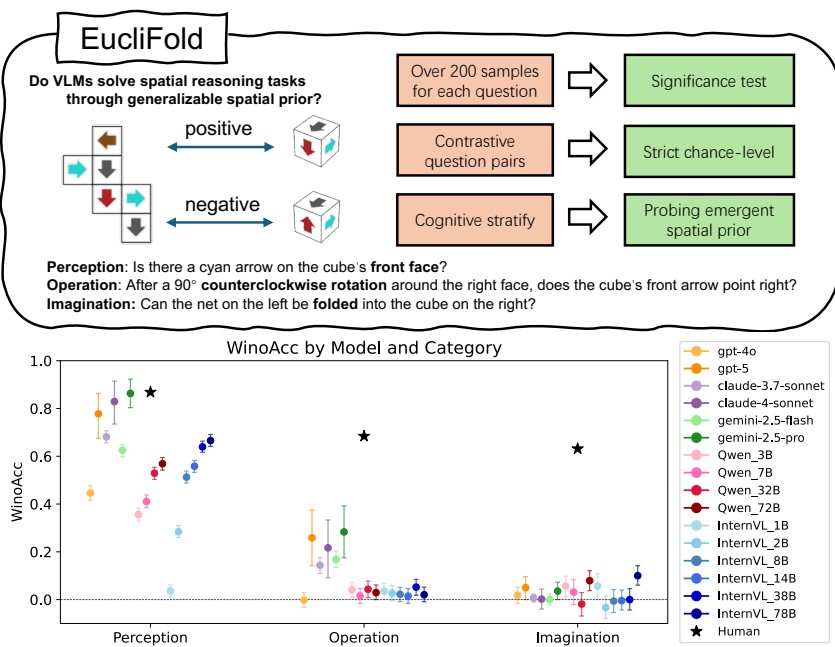

Figure 1: EucliFold design methodology and empirical findings. EucliFold (1) uses programmatic template-based generation to create sufficient samples for each question type, enabling statistical significance testing to confirm genuine above-chance performance; (2) employs paired samples with minimal contrastive differences to measure WinoAcc, thereby eliminating prompt bias and avoiding artificially low "below-chance" performance; and (3) adopts a cognitively stratified question set to prevent spurious above-chance performance driven by perception-level correlations. Empirical results validate this design methodology by faithfully distinguishing between non-ability and genuine above-chance ability. For numerical results, see Appendix F

For our empirical investigation, as shown in Figure 1, we focus on cube net folding tasks within Euclidean space and developed a synthetic dataset **EucliFold**. This choice is motivated by several key advantages: Euclidean representations provide mathematical precision for rigorous analysis; cube folding scales from basic perception to complex imagination-level reasoning; and systematic task variation enables controlled evaluation while maintaining sufficient complexity to reveal meaningful capability differences.

To address VLMs' intrinsic biases—including perception bias (Wang et al., 2024), pre-training bias (Lin et al., 2024), and response bias (Zheng et al., 2024)—we develop a bias-resistant evaluation methodology. We employ "Winograd-Style Accuracy" that compares performance on minimal-pair samples differing only in critical spatial content. This approach, inspired by the Winograd Schema Challenge (Levesque et al., 2012; Thrush et al., 2022), isolates genuine spatial reasoning from statistical artifacts by measuring the difference between true-belief and false-belief response patterns.

Our evaluation reveals that while current VLMs achieve reasonable performance on perceptual tasks, they struggle significantly with operational spatial reasoning and fail almost entirely at imagination-level tasks requiring autonomous spatial problem-solving. These findings suggest that current VLMs lack robust internal spatial representations and rely heavily on superficial pattern matching rather than genuine geometric understanding.

We make four contributions: (1) a systematic decomposition of spatial reasoning into distinct cognitive levels; (2) a controlled synthetic dataset EucliFold eliminating confounds while maintaining complexity; (3) bias-resistant evaluation distinguishing high-level Euclidean prior from low-level pattern matching; (4) comprehensive analysis revealing systematic VLM spatial reasoning gaps.

## 2 RELATED WORK

### 2.1 LARGE LANGUAGE MODEL-BASED VISUAL LANGUAGE MODELS

With the rapid development of large language models (LLMs) and the prohibitive cost of training large models from scratch, researchers increasingly build Visual Language Models (VLMs) by integrating visual encoders with pre-trained LLMs. This approach inherits world knowledge and reasoning capabilities from the underlying language model. Pioneer studies such as BLIP-2 (Li et al., 2023), LLaVA (Liu et al., 2023b), and MiniGPT-4 (Zhu et al., 2023) demonstrate significantly more robust instruction-following and broader zero-shot capabilities compared to previous train-from-scratch VLMs like CoCa (Yu et al., 2022). These advances have substantially expanded the scope of zero-shot visual question answering, catalyzing the development of comprehensive benchmarks such as MME (Zhang et al., 2021), MMMU (Yue et al., 2024), and MMBench (Liu et al., 2024) to evaluate VLM capabilities across diverse domains. Evaluations reveal that while current VLMs excel at OCR and visual grounding, they struggle with mathematical reasoning and real-world understanding (Chen et al., 2024; Bai et al., 2025). Spatial reasoning tasks particularly expose these limitations, requiring both geometric understanding and logical inference—two areas where current VLMs show systematic weaknesses.

### 2.2 SPATIAL REASONING BENCHMARKS FOR VLMS

While general-purpose VLM benchmarks like MMBench (Liu et al., 2024) contain spatial understanding tasks, several benchmarks specifically target VLM spatial capabilities. 3DSRBench (Ma et al., 2024) collects 2,772 human-annotated questions about 3D spatial reasoning, revealing that leading VLMs achieve only 50% accuracy compared to 90% human performance. LEGO-Puzzles (Tang et al., 2025) curates 1,100 questions from generated LEGO images, showing GPT-4o achieves 60% accuracy while humans reach 93.6%. VSI-Bench (Yang et al., 2025) evaluates VLM cognitive mapping with video inputs. Beyond evaluation, some benchmarks aim to improve spatial abilities through fine-tuning. Spatial Aptitude Training (SAT) (Ray et al., 2024) and Sparkle (Tang et al., 2024) demonstrate non-trivial performance gains but remain far from human-level performance. These findings suggest systematic limitations in VLM spatial reasoning. Our work probes a fundamental source of this deficiency: the lack of robust 3D Euclidean priors that enable flexible spatial understanding.

Table 1: Related work that evaluate spatial abilities of VLMs. MC: multi-choice; IT: interactive; TF: ture-false. Strict chance-level: a well-defined chance level below which weaker models do not fall.

| Related work | Data size | QA type | Avoid data contamination | Significance test | Strict chance-level | Avoid perception deficits |
|---|---|---|---|---|---|---|
| Ramakrishnan et al. (2024) | – | MC/IT | – | ✓ | | |
| Jia et al. (2025) | 8.4k | MC | – | | | |
| Wang et al. (2025) | 1.2k(inf) | MC | programmatic generation | | | |
| Li et al. (2025b) | 4k | MC/TF | – | | | |
| Li et al. (2025a) | 824+91 | MC | private test set | ✓ | | |
| Stogiannidis et al. (2025) | 1,800 | MC/TF | – | | | |
| Xu et al. (2025) | 312 | MC/TF | – | | | |
| This work | 8k(inf) | TF | programmatic generation | ✓ | ✓ | ✓ |

## 3 SPATIAL PRIORS OF THREE COGNITIVE LEVELS

Understanding spatial reasoning in artificial systems requires decomposing the underlying cognitive processes. Drawing from cognitive science research on spatial cognition and mental imagery (Shepard & Metzler, 1971), we propose that robust 3D spatial reasoning emerges from three hierarchical capabilities: **Perception**, **Operation**, and **Imagination**. Each level exhibits qualitatively distinct characteristics and builds upon the previous one to enable increasingly sophisticated spatial reasoning.

**Perception: Grounding Spatial Concepts.** At the foundational level, Perception encompasses grounding multimodal inputs into coherent internal spatial representations. The robustness of these internal spatial representations defines the perception-level spatial priors of humans or artificial models. Tasks such as spatial relation extraction (Liu et al., 2023a) evaluate the quality of perception-level spatial priors. *Generalizability* at this level stems from cross-modal consistency and distribution-invariant spatial relations that depend on relative geometric relationships rather than specific sensory modalities.

**Operation: Manipulating Spatial Representations.** Building upon perceptual grounding, Operation involves systematic manipulation of spatial representations according to external instructions or rules. This requires knowledge of spatial transformation dynamics and geometric mappings. Tasks such as dynamic prediction (Yi et al., 2020) evaluate the quality of operation-level spatial priors. *Generalizability* emerges from understanding predictable spatial dynamics that transfer across novel contexts. Crucially, operational competence provides the foundation for self-prediction when applying transformations—a prerequisite for imagination-level reasoning.

**Imagination: Autonomous Spatial Problem-Solving.** At the highest level, Imagination represents autonomous generation and evaluation of spatial operations under spatial constraints. *Generalizability* stems from internalizing fundamental geometric principles, enabling flexible reasoning over open-ended operation sets and novel problem configurations.

This hierarchical framework has critical implications for evaluation: systems may exhibit superficially impressive higher-level performance while relying on brittle, correlation-based strategies at lower levels. Such systems fail when encountering distribution shifts or requiring genuine geometric reasoning rather than pattern matching.

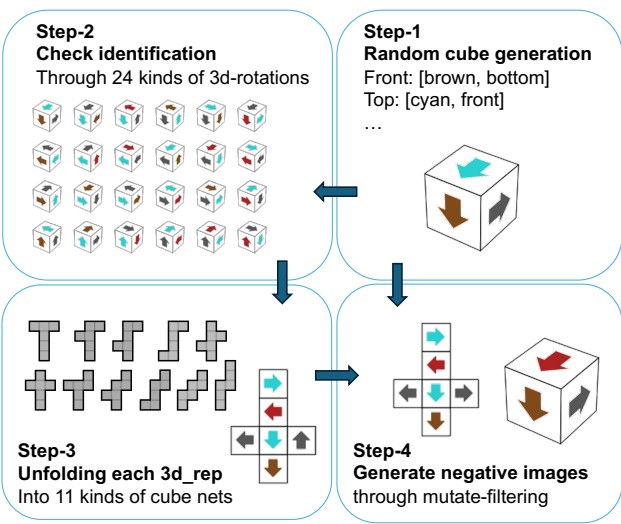

Figure 2: EucliFold Data Generation Process

# 4 EUCLIFOLD DATASET

## 4.1 SPATIAL REPRESENTATION DESIGN AND IMAGE GENERATION

The tasks in EucliFold center on cube net folding, chosen for three key properties: (1) it requires object manipulation and matching within three-dimensional Euclidean space, (2) it admits multiple valid spatial operation sequences without a fixed methodology, yet each solution pathway involves non-trivial reasoning, and (3) it maintains simplicity at both perceptual and cognitive levels without introducing excessive extraneous complexity.

Building upon the cube folding framework, we designed cube surface patterns that maintain diversity while avoiding excessive cognitive burden. Each cube face displays an arrow oriented parallel to the cube's edges (yielding four possible orientations), with each arrow rendered in one of eight possible colors (consistent with the established CLEVR benchmark Johnson et al. (2017)). Under this design, there exist at least $\frac{(8 \times 4)^6}{24} > 44,739,242$ distinct cubes and $(8 \times 4)^3 = 32,768$ different three-dimensional cube views.

To minimize spurious correlations at lower cognitive levels, EucliFold employs a systematic generation pipeline (Figure 2) that produces the required 3D cube views and 2D cube net images through uniform sampling based on symmetry principles at each stage. More details are in A

## 4.2 CONTRASTIVE QUESTION PAIRS

We employ fixed templates (see Appendix C) to test VLMs, varying only key vocabulary or image content. By minimizing differences between positive and negative samples, we control for VLMs' inherent biases, particularly response bias Zheng et al. (2024).

**Perception Tasks** All images in EucliFold are generated using the Python package *matplotlib*, which significantly differs from the image input distribution that most VLMs encounter during training. To ensure narrative precision, we use fixed text templates to describe problems, which also differs from everyday conversational language. This distributional shift poses challenges to VLMs' perceptual generalization capabilities.

Perception-level tasks examine whether VLMs can achieve robust spatial concept understanding under EucliFold's language-image distribution. We generate images $I(\text{3d\_rep})$, text $T(\text{3d\_rep})$, and incorrect text $T_{\text{neg}}(\text{3d\_rep})$ based on three-dimensional spatial representations, asking models to judge image-text matching.

Ideally, we test whether models can map both images and text to correct internal spatial concepts. Limited by interaction modalities, we can only estimate spatial grounding through image-text matching: $\textbf{Score}[I(\textbf{3d\_rep}), T(\textbf{3d\_rep})] \sim \textbf{Score}[I(\textbf{3d\_rep}), \textbf{3d\_rep}] \times \textbf{Score}[T(\textbf{3d\_rep}), \textbf{3d\_rep}]$

Through uniform sampling of *3d_rep*, we attempt to offset spurious matches based on irrelevant visual content and restore matches based on spatial concepts.

We design two perception tasks, *color recognition* and *orientation recognition*, to test whether models understand cube orientations in three-dimensional space and whether they understand the three-dimensional orientations corresponding to two-dimensional patterns on cube surfaces. The *color recognition* and *orientation task* uses the following statement template:

> *There is a **{color/wrong color}** arrow on the {visible face} face of the cube.*
> *The arrow on the {visible face} face of the cube is pointing towards the {orientation}.*

Since VLMs achieve extremely high accuracy in color recognition, we use color as an indicator to test models' perception and understanding of different cube surfaces. Building on cube surface perception, we can contrastively analyse the *orientation tasks* to judge whether models can correctly perceive the arrow directions on cube surfaces.

**Operation Tasks** Operation tasks examine whether VLMs can understand three-dimensional spatial rotations of cubes. While other spatial operations could be examined, we choose three-dimensional rotation to maintain consistency with the text-image distribution of perception problems.

We generate images $I(\text{3d\_rep})$ based on three-dimensional spatial representations, operation text $T_{\text{op}}(\text{3d\_op})$ based on three-dimensional rotation operations, and result text $T(\text{3d\_op}(\text{3d\_rep})) = T(\text{final\_3d\_rep})$ based on final representations. Negative samples are $T(\text{wrong\_final\_3d\_rep})$. We expect models to complete text-image matching in stages:

$$\begin{aligned} &\text{Score}[I(\text{3d\_rep}), T_{\text{op}}(\text{3d\_op}) + T(\text{3d\_op}(\text{3d\_rep}))] \\ &\sim \text{Score}[I(\text{3d\_rep}), \text{3d\_rep}] \times \text{Score}[T(\text{3d\_op}), \text{3d\_op}] \\ &\quad \times \text{Score}[\text{3d\_op}(\text{3d\_rep}), \text{final\_3d\_rep}] \times \text{Score}[T(\text{final\_3d\_rep}), \text{final\_3d\_rep}] \end{aligned} \tag{1}$$

The key operation is $\text{Score}[\text{3d\_op}(\text{3d\_rep}), \text{final\_3d\_rep}]$, measuring models' ability to perform operations based on spatial instructions. Through uniform sampling of *3d_op* and *3d_rep*, we aim to offset spurious correlations at the perceptual level $\text{Score}[I(\text{3d\_rep}), \text{3d\_rep}]$ and $\text{Score}[T(\text{final\_3d\_rep}), \text{final\_3d\_rep}]$, as well as spurious correlations based on specific linguistic configurations (e.g., based solely on $T_{\text{op}}(\text{3d\_op})$). This allows us to examine models' understanding of the **spatial operation concepts** $\text{Score}[T(\text{3d\_op}), \text{3d\_op}]$ and the dynamic functions of them $\text{Score}[\text{3d\_op}(\text{3d\_rep}), \text{final\_3d\_rep}]$. the statement of the **operation task** is as following:

> *If the cube rotates {90/270} degrees counterclockwise around its {visible face} face, the arrow on the initial {visible face} face of the cube will point to the **{orientation/wrong orientation}**.*

**Imagination Tasks** We design two imagination tasks: *folding* and *matching* to examine whether models can understand cubes and their nets as constant objects in three-dimensional space that maintain consistency after arbitrary reasonable transformations.

The folding task examines the correspondence between cubes and their nets. Under EucliFold's three-dimensional cube view settings, each net has 24 possible final folding configurations, and the folding actions to achieve each configuration are arbitrary. Our text provides no feasible folding action sets or traversal strategies.

The matching task examines whether two nets can be folded into identical cubes. This task also has many feasible spatial operation schemes, such as folding both nets separately and then performing rotational matching, or traversing all local adjacency relationships. The statements for folding and matching tasks, respectively:

> *The cube in Image-2 can be formed by folding the net shown in Image-1.*
> *The cube net in Image-1 and the cube net in Image-2 can be folded into identical cubes.*

We control image content to offset shallow perceptual correlations. For folding tasks, we generate reference net images $I_{\text{net}}(\text{2d\_rep}(\text{cube}))$ and correct folded images $I_{\text{cube}}(\text{3d\_rep}(\text{cube}))$. For negative samples, randomly different cubes have excessive differences that allow models to easily exclude negative samples. Therefore, we employ a mutate-filtering approach to select perceptually similar images that do not belong to the same cube: $I_{\text{cube\_neg}}(\text{3d\_rep}) = I_{\text{cube}}(\text{3d\_rep\_similar})$ where $\text{3d\_rep\_similar} \notin \text{3d\_rep\_set}(\text{cube})$.

Through uniform sampling of cube, *3d\_rep*, and *2d\_rep*, we aim to minimize the impact of spurious correlations at the perceptual level and examine models' ability for autonomous three-dimensional spatial matching.

$$\text{Score}[I_{\text{cube}}(\text{3d\_rep}(\text{cube})), I_{\text{net}}(\text{2d\_rep}(\text{cube}))]$$
$$\sim \text{Score}[I_{\text{cube}}, \text{cube}] \times \text{Score}[I_{\text{net}}, \text{cube}] \qquad (2)$$
$$\times \textbf{Score[cube\_spatial\_transformation]}$$

The matching task does not inherently require more qualitative abilities than the folding task but poses greater pressure on working memory. The folding task only requires traversal matching of three cube faces, while matching requires traversal matching of six faces.

### 4.3 WINOGRAD-STYLE ACCURACY

For VLM evaluation, besides designing task distributions to eliminate superficial spurious correlations, we must also eliminate response bias Zheng et al. (2024). Specifically, VLM answer tokens True/False are influenced by the joint attention of all preceding tokens, many of which are unnecessary. These specific irrelevant contexts create strong tendencies toward True or False tokens.

To minimize the impact of such tendencies on measurement results, we adopt and adjust the metric from Winoground Thrush et al. (2022). We measure models' relative beliefs under specific contexts through answer differences between paired positive and negative samples with minimal necessary differences.

Specifically, for each pair of questions (pos, neg), we count two types of answer combinations:

- **True-belief:** (correct, correct), or equivalently, (True positive, True negative)
- **False-belief:** (incorrect, incorrect), or equivalently, (False negative, False positive)

The remaining two answer combinations cannot reflect the differences between paired questions, indicating that VLMs cannot effectively distinguish between them. We use the difference between true-belief and false-belief to measure model capability, termed:

$$\text{Winograd-Style Accuracy} = \mathbb{P}[\text{true\_belief}] - \mathbb{P}[\text{false\_belief}] \qquad (3)$$

For confidence interval calculation, we assume $\mathbb{P}[\text{true\_belief}]$ and $\mathbb{P}[\text{false\_belief}]$ are independent. We calculate $\text{VAR} = \hat{p} \times (1 - \hat{p})/n$ separately, thus $\text{VAR}(\text{Winograd-Style Accuracy}) = \text{VAR}(\text{true\_belief}) + \text{VAR}(\text{false\_belief})$. Finally, we use normal distribution approximation Z-scores to obtain confidence intervals.

## 5 EXPERIMENTS AND ANALYSIS

**Model Selection.** We choose GPT-4o (Hurst et al., 2024), Gemini-2.5-flash (Team et al., 2023), and Claude-3.5-sonnet (Anthropic, 2024) as representative high-performance closed-source VLMs. We select the Qwen-VL-2.5 series (Bai et al., 2025) as representative general-purpose VLMs and the InternVL-2.5 series (Chen et al., 2024) as representative post-trained visual reasoners.

**Experimental Settings.** For each model, we employ system prompts to control chain-of-thought reasoning (Kojima et al., 2022), followed by true-false answers. More specific parameter configurations are detailed in Appendix D. Each datapoint in this section represent around 1,200 samples, for detail, see Appendix B.

### 5.1 GENERAL PERFORMANCE AND THE EFFECT OF SCALING

As shown in Figure 1, overall, most of the open-source and closed-source models demonstrate relatively high score at the perception level, though gaps remain compared to human performance. At the operation level, only Gemini and Claude show performance significantly above chance level, yet still far weaker than humans. At the imagination level, only InternVL-78B and Qwen-72B perform significantly above chance level, still substantially below human performance.

**Perception Tasks.** The color recognition task at the perception level primarily judges whether models can distinguish different faces of cubes. We ask models about the color of specific cube faces (e.g., top face arrow color) and set negative colors from the other two visible faces as distractors. As shown in Figure 3, except for extremely small models, most achieve near-perfect performance.

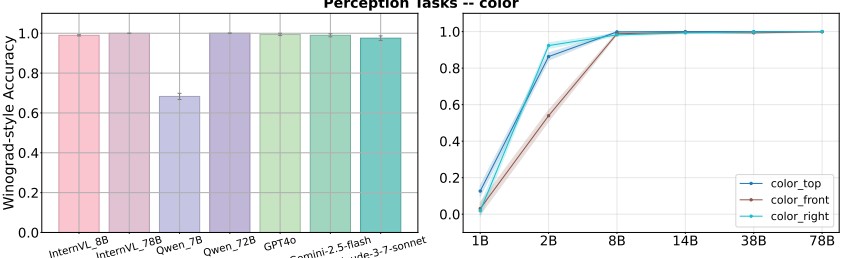

Figure 3: Performance scaling on color recognition tasks across different model sizes.

The arrow orientation recognition task primarily judges whether models can understand the orientation of two-dimensional objects in three-dimensional space. As shown in Figure 4, accuracy shows an increasing trend with model scale but does not reach perfect levels. This may be due to VLMs' insufficient accuracy in perceiving fine-grained content in images (Fu et al., 2024), or lack of internal three-dimensional spatial concepts, preventing proper grounding of image content to sufficiently discriminative three-dimensional spatial representations.

On the other hand, models show significant differences in perception accuracy across different cube faces. Since color task accuracy saturates at 8B parameters, indicating that models can perfectly distinguish the three faces, the orientation perception accuracy differences may be due to varying distortions of the arrows, resulting in different perception difficulties.

### 5.1.1 OPERATION TASKS: PROMPTING

As illustrated in Figure 5, only two close-source models (Gemini-2.5-flash and Claude-3.7-sonnet) perform above chance-level. The performance gap between perception and operation tasks indicates that while models can recognize static spatial configurations to some extent, they struggle with dynamic spatial transformations.

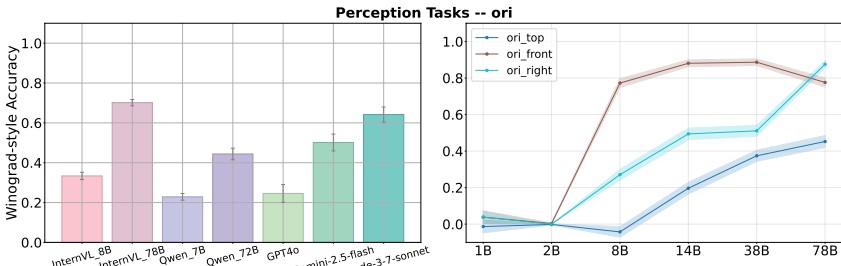

Figure 4: Performance scaling of the InternVL-2.5 family on orientation recognition tasks across different model sizes and cube faces.

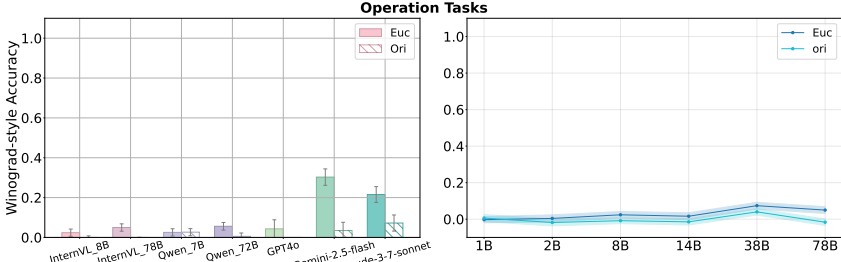

Figure 5: Comparison of direct answer vs. chain-of-thought prompting on folding tasks.

**Ablation Study on Prompting.** Parallel to orientation tasks, we change the description style to Euclidean terms, forming another set of questions (Euc in short, for detail see Appendix E The results in Figure 5 indicate a significant performance gain when shifting from natural-style language (ori) to mathematical style (Euc). Although we implement multiple strategies to avoid various biases in VLM evaluation, we cannot fully control prompt-induced biases in VLMs' internal chain-of-thought preferences.

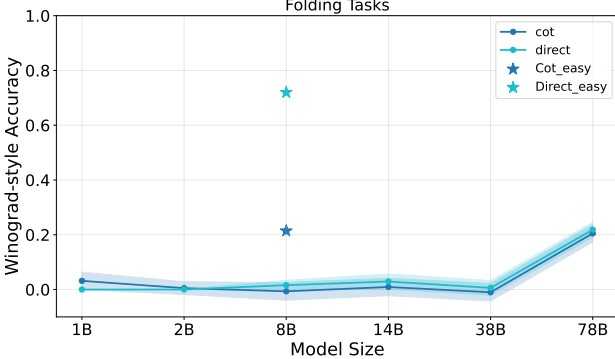

Figure 6: Ablation study on folding tasks: direct vs. chain-of-thought prompting and leaked vs. proper negative samples.

### 5.1.2 IMAGINATION TASKS: FAILURE IN COMPLEX SPATIAL REASONING

As shown in Figure 1, folding tasks show that only the InternVL series can exceed chance level, possibly benefiting from similar tasks in InternVL's post-training process that enable solving partial problems.

As mentioned earlier, InternVL's folding ability may stem from learning a specific non-generalizable strategy rather than possessing genuine generalizable spatial imagination capabilities. We verify this hypothesis through two ablation studies.

**Ablation Study on Chain-of-Thought Prompting** We test InternVL's accuracy on folding tasks using direct prompting. Since generalizable spatial folding strategies must involve multi-step traversal and enumeration, direct-answer approaches cannot encode such variable-length strategies (Merrill

et al., 2022). Therefore, as shown in Figure 6, when direct-answer format approaches chain-of-thought accuracy, it must be utilizing a fixed strategy to complete partial spatial matching.

**Ablation Study on easy negative samples** We design leaked negative samples for folding tasks, where negative cube samples contain colors not present in the net, allowing models to use simple strategies to judge negative samples (comparing overall color sets between cube and net). Results show that even 8B models achieve accuracy significantly above chance level on leaked negative sample settings.

These two ablation experiments demonstrate that InternVL lacks imagination-level Euclidean priors and emphasize the necessity of precisely controlling negative samples.

## 6 CONCLUSION AND DISCUSSION

Our evaluation reveals fundamental limitations in current VLMs' spatial reasoning capabilities. While state-of-the-art models demonstrate reasonable performance at the Perception level, they fail dramatically at Operation and Imagination levels, the best model achieving only 0.28% and 0.10% accuracy respectively. This stark capability gap suggests that VLMs rely primarily on statistical pattern matching rather than developing genuine geometric understanding of 3D transformations. Our cognitively-stratified framework effectively isolates these different levels of capability, revealing that spatial reasoning deficits are not uniform but concentrated in higher-order operations requiring mental manipulation of spatial representations. These findings align with cognitive neuroscience research that spatial pirors are mainly on distinct neural circuits (O'Keefe & Dostrovsky, 1971; Moser et al., 2008), while challenging the assumption that scaling data and parameters alone will bridge the human-AI gap in spatial reasoning.

This work contributes EucliFold, a cognitively-inspired benchmark for evaluating 3D spatial reasoning in VLMs, along with a bias-resistant evaluation methodology that minimizes confounding factors. Our three-level decomposition framework offers a principled approach to capability assessment that could be adapted to other cognitive domains. The systematic nature of current failures across different model architectures suggests that achieving human-level spatial intelligence may require architectural innovations or training paradigms that explicitly incorporate geometric inductive biases rather than incremental improvements to existing approaches. While our cube net domain provides rigorous controlled evaluation, future work should investigate generalization to other spatial reasoning tasks and explore whether explicit geometric training can address the fundamental limitations we identify. Our methodology demonstrates how insights from cognitive science can inform AI evaluation, potentially leading to more robust benchmarks for assessing genuine reasoning capabilities beyond pattern matching.

### AUTHOR CONTRIBUTIONS

If you'd like to, you may include a section for author contributions as is done in many journals. This is optional and at the discretion of the authors.

### ACKNOWLEDGMENTS

Use unnumbered third level headings for the acknowledgments. All acknowledgments, including those to funding agencies, go at the end of the paper.

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

## A APPENDIX: DATA GENERATION PIPELINE DETAILS

There are 5 steps to generate all configurations and images.

**(0) Spatial representation design.** We establish a three-dimensional representation (*3d_rep*) and two-dimensional representation (*2d_rep*) to encode cube spatial states and net configurations, respectively.

**(1) Distinct cube generation.** We randomly generate arrow patterns on cube surfaces, then apply the 24 rotational group transformations of *3d_rep* to produce 24 equivalent representations. After comparing against existing cube representations and filtering duplicates, we obtain a collection of distinct cubes.

**(2) Positive 3D view generation.** Each cube corresponds to 24 possible *3d_rep* configurations. Since only three faces are visible in any view, we filter cubes that would produce duplicate 3D view images.

**(3) Positive 2D net generation.** Each 3D-rotational variant of a cube corresponds to 11 distinct two-dimensional nets (excluding 2D-rotational and 2D-mirror symmetries), yielding $24 \times 11 = 264$ possible net configurations per cube.

**(4) Negative sample generation.** To produce sufficiently challenging negative samples, we employ a *mutation-validation* approach rather than random generation, ensuring that negative images maintain plausible appearance similar to positive images while violating geometric constraints. The detailed methodology is presented in the following section.

## B APPENDIX: DATA GENERATION PIPELINE DETAILS

For data generation, we first create 50 distinct cubes, then generate three-dimensional cube views for each of the 24 three-dimensional representations per cube, yielding 1,200 cube images in total. Upon inspection, the image duplication rate is below 5%. For each image, we generate corresponding positive and negative samples for perception and operation questions, resulting in 1,200 pairs per question type.

For folding questions, we randomly rotate each cube image, then randomly select a method to unfold it, obtaining a net image as reference. We then randomly mutate the color or shape of one face to generate a negative cube image, thus obtaining paired samples. For matching questions, we randomly select 24 nets for each cube, then perform mutations, also yielding 1,200 question pairs.

## C APPENDIX: QUESTION TEMPLATES

Template for one-image tasks:

> **Image:** {image of a 3D cube view}
> **Question:** *Based on the image and the description of the image, is the following statement True or False?*
> **Description:** *The image shows a cube with three visible faces (top, front, right), each face has an arrow on it.*
> **Statement:** {question-specific statement}
> **Answer:** {possible chain-of-thought} **{True or False}**

Template for two-images tasks:

> **Image-1:** {image-1}
> **Image-2:** {image-2}
> **Question:** *Based on Image-1 and Image-2, is the following statement True or False?*
> **Statement:** {question-specific statement}
> **Answer:** {possible chain-of-thought} **{True or False}**

## D APPENDIX: VLMS EVALUATION CONFIGURATIONS

**System Prompt Settings.** As for the main experiment, we use the same chain-of-though style System Prompt for all models. The *perception* and *operation* tasks contains one image, *imagination* tasks contains two images.

System Prompt for *chain-of-though* setting

> The following is a True/False question based on **{an image/two images}**. Analyze the image and the question carefully, then determine if the statement is True or False. Provide your reasoning step by step.

System Prompt for *direct-answer* setting

> The following is a True/False question based on **{an image/two images}**. Directly output only 'True' or 'False' as your answer. Do not provide any reasoning, explanation, or additional text.

System Prompt for *short-style* setting

> The following is a True/False question based on **{an image/two images}.**

System Prompt for *base-style* setting

> You are a helpful assistant.

**Chain-of-Thought Settings.** We use temperature=0.0 for chain-of-thought text generation and temperature=1.0 for direct text generation.

## E    APPENDIX: EUCLIDEAN STYLE QUESTIONS

Templates for Euclidean style questions. Statement:

> *Imagine that the cube is set at the center of a 3D space, and let the positive x, y, and z axes point out from the right, top, and front faces, respectively.*

Perception question template:

> *The arrow on the {**right**} face of the cube is pointing in the {**positive-x**} direction.*

Operation question template:

> *If the cube rotates {90/270} degrees counterclockwise around around the {**positive-y**} axis, the arrow on the initial {visible face} face of the cube will point to the {**negative-z**}.*

## F    APPENDIX: COMPLETE RESULTS

Table 2: WinoAcc and confidence interval (95%) on Perception, Operation, Folding, and Matching questions.

| Model | Perception | Operation | Folding | Matching |
|---|---|---|---|---|
| Chance | 0.000 | 0.000 | 0.000 | 0.000 |
| gpt-4o | 0.446 [0.416, 0.477] | -0.002 [-0.033, 0.029] | 0.008 [-0.058, 0.075] | 0.029 [**0.000**, 0.054] |
| gpt-5 | 0.778 [0.675, 0.863] | 0.258 [0.142, 0.375] | 0.092 [0.013, 0.167] | 0.008 [-0.042, 0.058] |
| claude-3.7-sonnet | 0.681 [0.656, 0.707] | 0.144 [0.110, 0.176] | 0.013 [-0.017, 0.050] | 0.000 [**0.000**, 0.000] |
| claude-4-sonnet | **0.829** [0.735, 0.914] | 0.217 [0.092, 0.333] | 0.017 [-0.071, 0.096] | -0.013 [-0.042, 0.021] |
| gemini-2.5-flash | 0.625 [0.598, 0.650] | 0.169 [0.134, 0.203] | -0.004 [-0.050, 0.037] | 0.004 [**0.000**, 0.013] |
| gemini-2.5-pro | 0.863 [0.803, 0.923] | **0.283** [0.175, 0.392] | 0.037 [0.000, 0.075] | 0.033 [-0.033, 0.100] |
| Qwen_3B | 0.356 [0.328, 0.384] | 0.041 [0.010, 0.072] | 0.067 [0.004, 0.133] | 0.046 [-0.008, 0.100] |
| Qwen_7B | 0.410 [0.385, 0.438] | 0.015 [-0.017, 0.046] | 0.013 [-0.071, 0.092] | 0.050 [-0.008, 0.108] |
| Qwen_32B | 0.529 [0.503, 0.554] | 0.043 [0.008, 0.077] | -0.013 [-0.092, 0.067] | -0.025 [-0.096, 0.042] |
| Qwen_72B | 0.569 [0.542, 0.595] | 0.029 [0.000, 0.061] | 0.146 [0.062, 0.229] | 0.013 [-0.004, 0.033] |
| InternVL_1B | 0.036 [0.012, 0.061] | 0.035 [0.003, 0.068] | 0.079 [-0.000, 0.163] | 0.033 [-0.046, 0.117] |
| InternVL_2B | 0.284 [0.259, 0.310] | 0.027 [-0.005, 0.060] | -0.025 [-0.083, 0.025] | -0.042 [-0.125, 0.033] |
| InternVL_8B | 0.513 [0.488, 0.538] | 0.022 [-0.009, 0.051] | 0.004 [-0.075, 0.079] | -0.017 [-0.062, 0.029] |
| InternVL_14B | 0.558 [0.533, 0.582] | 0.015 [-0.016, 0.046] | -0.004 [-0.079, 0.071] | -0.004 [-0.033, 0.025] |
| InternVL_38B | 0.639 [0.617, 0.663] | 0.052 [0.018, 0.085] | 0.004 [-0.079, 0.087] | -0.004 [-0.025, 0.013] |
| InternVL_78B | 0.666 [0.641, 0.691] | 0.020 [-0.009, 0.052] | **0.196** [0.117, 0.271] | 0.004 [**0.000**, 0.013] |
| Human | 0.868 | 0.684 | 0.816 | 0.447 |

## G    APPENDIX: HUMAN PERFORMANCE

Table 3: Human performance

| | question (pos+neg) | Positive Acc | Negative Acc | WinoAcc |
|---|---|---|---|---|
| perception | 76 (46+30) | 95.6 (44/46) | 90.0 (27/30) | 0.868 |
| operation | 76 (33+43) | 72.7 (24/33) | 93.0 (40/43) | 0.684 |
| folding | 76 (44+32) | 86.4 (38/44) | 96.9 (31/32) | 0.816 |
| matching | 76 (36+40) | 69.4 (25/36) | 75.0 (30/40) | 0.447 |

We sample 60 examples each from perception, operation, folding, and matching tasks, totaling 240 samples. We constructed a visualization webpage for testing, selecting university student volunteers as subjects. Each subject randomly answered 16 questions (4 from each category), and only complete responses were considered valid. We received 19 valid response batches, totaling 304 questions. The final answered questions distinctly cover 54 perception, 50 operation, 54 folding, and 50 matching questions.

For human subjects, seeing paired questions simultaneously provides additional useful information, so we do not use paired questions to test humans, assuming humans have no response bias. We estimate corresponding Winograd-Style Accuracy through human positive accuracy and negative accuracy:

$$\mathbb{P}[\text{true\_belief}] \sim \mathbb{P}[\text{pos\_acc}] \times \mathbb{P}[\text{neg\_acc}]$$
$$\mathbb{P}[\text{false\_belief}] \sim (1 - \mathbb{P}[\text{pos\_acc}]) \times (1 - \mathbb{P}[\text{neg\_acc}]) \quad (4)$$
$$\text{Human Winograd-Style Accuracy} = \mathbb{P}[\text{true\_belief}] - \mathbb{P}[\text{false\_belief}]$$

# H APPENDIX: CASE STUDY ON THE PARTIAL SOLUTION OF FOLDING TASK

For the folding task, InternVL-2.5 achieves the highest WinoAcc, which is also significantly above chance level. We first examined the distribution of its answers and then zoomed in to find that the all-correct question pairs suggest the model is adopting a partial solution that works only for certain configurations.

The statistics for all question pairs of InternVL-2.5 on the folding task are as follows: 719 all-correct pairs, 264 all-wrong pairs, 579 all-positive pairs, and 838 all-negative pairs. The asymmetry between the all-correct and all-wrong pairs leads to an above-chance WinoAcc, so we further analyze what kinds of question pairs the model gets correct. We find that 513 out of the 719 all-correct pairs involve an easy "L"-shaped pattern on the cube net (examples are shown below). If we remove these samples, the remaining 206 all-correct pairs become comparable in number to the 264 all-wrong pairs, resulting in a chance-level WinoAcc. This indicates that InternVL-2.5 attains its above-chance performance by adopting a partial solution that only handles specific cases in the folding task.

Here are two cases showing "L" shape and "non-L" shape:

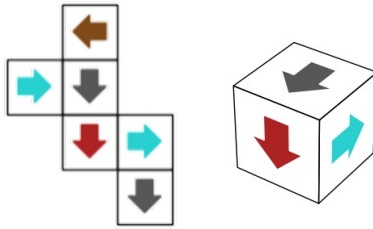

Figure 7: Good case ("L" shape) for Intern-VL-2.5

---

To determine if the cube in Image-2 can be formed by folding the net shown in Image-1, we need to analyze the arrangement of the arrows on both the net and the cube.

1. **Identify the Faces on the Net (Image-1):** - The net consists of six squares, each with an arrow.
- The arrows are as follows:
- Top face: Brown left arrow.
- Middle left face: Cyan right arrow.
- Middle center face: Gray down arrow.
- Middle right face: Red down arrow.
- Bottom left face: Cyan right arrow.
- Bottom center face: Gray down arrow.

2. **Identify the Faces on the Cube (Image-2):** - The cube shows three visible faces with arrows:
- Top face: Gray down arrow.
- Front face: Red down arrow.
- Right face: Cyan right arrow.

3. **Match the Faces:** - The top face of the cube (gray down arrow) matches one of the gray down arrows on the net.
- The front face of the cube (red down arrow) matches the red down arrow on the net.
- The right face of the cube (cyan right arrow) matches one of the cyan right arrows on the net.

4. **Check for Consistency:** - When folding the net into a cube, the relative positions of the arrows must match.
- The net shows that the gray down arrow (middle center) is adjacent to the red down arrow (middle right) and the cyan right arrow (middle left).
- On the cube, the gray down arrow (top face) is adjacent to the red down arrow (front face) and the cyan right arrow (right face).

5. **Conclusion:** - The arrangement of arrows on the net and the cube is consistent.
- Therefore, the cube in Image-2 can indeed be formed by folding the net shown in Image-1.

Answer: True

---

From the above case, we can see that the top, front, and right faces of the cube happen to form an "L"-shaped pattern in the net, without any rotation. In this situation, a simple pattern-matching strategy is sufficient to produce the correct positive answer.

When there is no direct L-shaped mapping, the model can make several types of errors. In the case below, it incorrectly treats one face as fixed in the net and ignores possible rotations. This leads to a false negative answer.

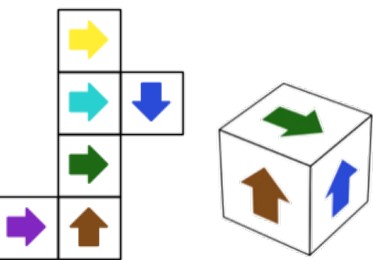

Figure 8: bad case (non-L shape) for Intern-VL-2.5

To determine if the cube in Image-2 can be formed by folding the net shown in Image-1, we need to compare the arrangement of the arrows on both images.

1. **Identify the Faces in Image-1:**
- Top face: Yellow arrow pointing right.
- Front face: Cyan arrow pointing right.
- Right face: Blue arrow pointing down.
- Bottom face: Green arrow pointing right.
- Left face: Purple arrow pointing right.
- Back face: Brown arrow pointing up.

2. **Identify the Faces in Image-2:**
- Top face: Green arrow pointing right.
- Front face: Brown arrow pointing up.
- Right face: Blue arrow pointing up.

3. **Compare the Faces:**
- In Image-1, **the green arrow is on the bottom face**, but in Image-2, the green arrow is on the top face. This discrepancy indicates that the net in Image-1 cannot be folded to match the cube in Image-2.

Based on this analysis, the statement is **False**. The cube in Image-2 cannot be formed by folding the net shown in Image-1.

# I    APPENDIX: LLM USAGE

Large Language Models (LLMs) were used as auxiliary tools for code refinement and text polishing. All LLM-generated content, including code and written text, was rigorously reviewed and validated by at least one author.

