# OpenReview forum: "EucliFold: Probing 3D Euclidean Prior in VLMs via Cognitively-Stratified Folding Tasks"
_ICLR.cc/2026/Conference — Submitted to ICLR 2026_

### Official Review · Reviewer_tg8N · 2025-10-31

**Soundness:** 2
**Presentation:** 2
**Contribution:** 2
**Rating:** 2
**Confidence:** 4

**Summary:**

This paper introduces a new benchmark called Euclifold to test Vision Language Models (VLMs) spatial understanding ability. The authors consider the tasks of both interpreting and manipulating synthetically rendered 3D cubes with colored arrows rendered on each face to generate diverse stimuli and associated queries that test physical reasoning.

Three different tasks of increasing complexity are introduced. The first task, Perception involves making predictions about the spatial structure of the cube, i.e. directions and colors of the arrows printed on each face of the cube. The second task, Operation involves understanding what the cube will look like when acted upon (i.e rotated by a certain angle). And finally, the third task, Imagination tests whether models can understand whether an unfolded cube (i.e net) can be folded back into it’s original configuration, or whether two unfolded cubes can be folded into the same cube.

The conclusion is that existing models are fine for perception, but fail on operation and imagination.

**Strengths:**

1. It’s an open question whether VLMs truly have 3D spatial understanding capabilities. There have been some benchmarks that test for this on natural images such as [1], but the full extent of the limitations is not understood yet. This benchmark is a good addition to the existing tests.
2. Introducing a hierarchy of tasks of increasing complexity is nice because it helps us make judgements about where exactly do models struggle (i.e perception or reasoning).
3. I appreciate that the authors tested scaling behavior. It’s important to test a wide range of model architectures before making any confident conclusions. It was interesting to see that for the folding task, we need to go to 80B to get above chance results, and even that is not really solving the problem. I agree with the authors that this challenges the assumption that scaling data and parameters alone will bridge the human-AI gap in spatial reasoning, especially given that the stimuli are simple and tasks are some of the easier 3D tasks in the range of 3D spatial reasoning capabilities that humans have.

**Weaknesses:**

1. I think the benchmark mainly functions as an adversarial test. It’s clear that models can’t solve this task, but does the benchmark help generate results which suggest a way forward?  The paper would benefit from a discussion of possible solutions. Otherwise, I’m not sure what’s to be gained from simply knowing that “model’s can’t do X”
2. The stimuli are also purely synthetic, and I’m not sure what would be the real world analog of this task. I get that it’s something that humans can do which models can’t and it’s worth understanding why, but what really is the practical relevance? Why should we care about cube folding? I think some arguments connecting these abstract reasoning tests to real-world spatial cognition, robotics, or embodied perception, or maybe even including some real world stimuli might help.
3. The paper evaluates only a few VLMs. It would be a good idea to consider evaluating a wider range of models to know for sure that all models struggle on these tasks.
4. There’s some prior work ( Unfolding Spatial Cognition [2] ) that also explored the cube folding task. It would strengthen the paper to clarify how Euclifold differs or improves on those benchmarks.

Writing Improvements:

L440: “achieving only X% and Y%” seems to be a typo.

The term “net” is abruptly introduced. It might be a good idea to define it clearly when first mentioned by adding a short visual or textual explanation of what a cube net is. It wasn’t obvious to me what cube nets are when I first read the paper.

Section 4.1.2 is titled “Imagination Task” but doesn't seem to have results for matching?

References
[1] Why Is Spatial Reasoning Hard for VLMs? An Attention Mechanism Perspective on Focus Areas
[2] Unfolding Spatial Cognition: Evaluating Multimodal Models on Visual Simulations

**Questions:**

1. What mechanisms might help VLMs move beyond pattern-matching toward true spatial reasoning? Does the benchmark help provide any insights on that?
2. Could this framework generalize to other shapes or real 3D object data?
3. How does Euclifold substantively differ from the Unfolding Spatial Cognition paper?

---

> ### Author Response · Authors · 2025-12-04
>
> Thanks for your constructive comments. The feedback has been invaluable in helping us clarify our contributions and improve the manuscript. We first provide a general response, and then address each weakness and question in detail.
>
> **General response**
> The motivation of EucliFold is fundamentally different from that of comprehensive spatial reasoning datasets, which primarily aim to measure overall spatial reasoning performance and to provide diverse, real-world-aligned samples for applications. By contrast, EucliFold is a diagnostic dataset designed to probe to what degree current VLMs learn (emergent) generalizable solutions for spatial reasoning.
>
> Designing a trustworthy metric of “emergence” is non-trivial. Such a metric should:
>
> 1. Provide a strict chance-level baseline and confidence intervals to quantify the probability that any above-chance performance is genuine.
> 2. Minimize shortcut biases to avoid “fake emergence” driven by spurious correlations.
> 3. Distinguish different levels of emergence (e.g., narrow partial solutions versus broadly generalizable solutions).
>
> Although EucliFold focuses only on the cube net folding task (a subset of tasks in some broader datasets), it is explicitly structured to meet these goals as closely as possible:
>
> 1. Structured question generation enables a WinoAcc-style evaluation with a well-defined chance level, ensuring that weaker models do not appear below chance simply due to dataset imbalance.
> 2. Over 128 contrastive pairs per question type provide narrow confidence intervals and high statistical power for detecting truly “no-emergence” behavior.
> 3. Cognitively stratified question levels share the same perceptual distribution, revealing that current SOTA models often learn only partial solutions rather than fully generalizable 3D Euclidean reasoning.

---

> ### Author Response · Authors · 2025-12-04
>
> **W1: Does the benchmark help generate results that suggest a way forward?**
>
> Response: We believe the answer is yes. While many spatial reasoning datasets (including EucliFold) show that current SOTA VLMs are weak at 3D spatial reasoning, EucliFold yields more specific guidance:
>
> We find that many large VLMs exhibit reasonably generalizable 3D perception (e.g., recognizing symbolic 3D configurations) but struggle with 3D operations (e.g., spatial transformations, composition of folds), whereas smaller models often fail even at the perceptual level.
> This pattern suggests a separation between perceptual grounding and geometric manipulation, and it aligns with prior studies on toy models [citation].
> Consequently, future work might focus on targeted training or architectural changes that strengthen the geometric reasoning component, rather than only scaling perceptual data.
>
> **W2: What would be the real-world analog of this task?**
>
> Response: Biologically, humans and many animals (including early mammals and even some reptiles) rely on the hippocampus as a functional spatial prior. Some studies have shown that AI models can develop grid-cell-like representations and implicit cognitive maps, but it remains unclear whether current SOTA VLMs have learned and can exploit similarly structured implicit priors.
>
> We hypothesize that if VLMs were able to utilize an implicit Euclidean spatial prior—analogous to human spatial cognition and hippocampal coding—they could solve many out-of-distribution spatial tasks, rather than handling each spatial scenario in an ad hoc, pattern-matching manner. This would move them closer to Artificial General Intelligence in the spatial domain.
>
> EucliFold aims to evaluate precisely this possibility: whether models rely on general Euclidean structure or merely on case-by-case pattern recognition. By stressing a controlled but difficult spatial task (cube net folding), EucliFold seeks to encourage and measure general spatial abilities, rather than just “versatile but non-general” skills.
>
> **W3: The paper evaluates only a few VLMs**
>
> Response: We have evaluated 13 models in total (10 open-source and 3 closed-source). In the revision, we further include 3 of the most recent SOTA models (GPT-5, Claude-4-Sonnet, and Gemini-2.5-Pro) to strengthen our empirical findings.
>
> Note: We initially experimented with the Gemini-3-pro-preview API, but encountered several bugs affecting “thinking mode” control. Because of these issues, we decided not to include that model to avoid an unfair comparison.

---

> ### Author Response · Authors · 2025-12-04
>
> **W4 & Q3: Relation to prior work on cube folding (Unfolding Spatial Cognition / STARE)**
>
> Response: The core motivational difference from “Unfolding Spatial Cognition” is summarized in the General Response. Here we briefly describe STARE and then highlight substantive differences.
>
> The Unfolding Spatial Cognition paper introduces STARE (Spatial Transformations and Reasoning Evaluation), a comprehensive dataset with ~4k samples covering 6 types of spatial tasks, curated from 6 diverse sources. STARE is designed to:
>
>     Cover a broad range of perceptual conditions (2D vs. 3D, simulated vs. real),
>     Include diverse spatial transformations (temporal relations, sequential assembly, spatial positioning),
>     Span different levels of ambiguity (implicit vs. explicit reasoning).
>
> While STARE provides rich, diverse test cases to show that VLMs struggle with spatial reasoning, EucliFold focuses on a single pivot task with structured questions and explicit quantitative controls, leading to several additional insights:
>
> **1. Clear chance levels and confidence intervals for near-zero emergent ability.**
>
> In STARE’s cube net folding subset, some models achieve accuracy substantially below the nominal chance level (50% for True/False). For example (excerpt from their Cube Net results):
>
> | Model                   | Cube Net ✗VSim | Cube Net ✓VSim |
> |-------------------------|----------------|----------------|
> | Random                  | 50.5           | 50.5           |
> | GPT-4o                  | 50.3           | 52.2 (↑ 1.9)   |
> | Claude-3.5 Sonnet       | 52.3           | 51.6 (↓ 0.7)   |
> | Gemini-2.0 Flash        | **37.7**           | 35.6 (↓ 2.1)   |
> | Gemini-2.0 Flash Think  | 48.3           | 50.7 (↑ 2.4)   |
> | o1                      | 51.3           | 53.4 (↑ 2.1)   |
> | LLaVA-OneVision-72B     | **28.5**           | 34.2 (↑ 3.7)   |
> | InternVL2.5-78B         | 37.1           | 37.3 (↑ 0.2)   |
> | Qwen2.5-VL-3B           | 43.5           | 41.0 (↓ 2.5)   |
> | Qwen2.5-VL-7B           | 40.7           | 44.9 (↑ 4.2)   |
> | Qwen2.5-VL-72B          | **35.2**           | 53.4 (↑ 18.2)  |
>
> Such below-chance performance suggests that if we systematically flipped the True/False labels, some models would appear “smart” simply by exploiting consistent biases. In other words, **systematic bias can create fake cleverness or fake stupidity**.
>
> EucliFold’s synthetic framework is built to avoid this issue: it enforces balanced contrastive pairs and uses WinoAcc-style evaluation with confidence intervals, yielding a strictly controlled chance baseline and allowing robust detection of genuine vs. spurious emergent behavior.
>
> **2. Different empirical findings and diagnostic insights.**
>
> The Unfolding Spatial Cognition paper offers a detailed cube folding analysis but primarily at the level of whether faces are foldable. They observe “low accuracy in correctly identifying folded faces (57.4%)”, but this alone does not distinguish whether failures arise from:
>
>   Misperception of 3D configurations, or
>   Inability to perform the required 3D reasoning given correct perception.
>
> EucliFold, by design, separates perception-level tasks (e.g., identifying symbols or visible faces) from imagination/operation-level tasks (e.g., mentally folding or transforming the cube), all under the same perceptual distribution. This allows us to localize failure modes more precisely—for example, showing that many models have adequate 3D perception but fail at the imagination/operation stage, indicating a lack of robust Euclidean priors rather than purely visual deficits.

---

> ### Author Response · Authors · 2025-12-04
>
> **Q1: What mechanisms might help VLMs move beyond pattern matching toward true spatial reasoning?**
>
> Response: EucliFold suggests at least two promising directions:
>
> **1. Pre-training on 3D spatial reasoning data (rather than only post-training).**
>
> Our results indicate that current VLMs often learn partial solutions that exploit shortcuts, implying that the underlying LLM backbone does not reason with a coherent 3D spatial prior. Incorporating structured 3D spatial reasoning data at the pre-training stage may encourage the model to internalize more generalizable Euclidean structure.
>
> **2. Integrating external simulation tools for post-training.**
>
> Another avenue is to couple VLMs with 3D simulation or physics engines and train them (e.g., via RL or tool-augmented learning) to plan and verify spatial transformations. This could help models move from pattern-matching to explicitly grounded spatial reasoning.
>
> **Q2: Could this framework generalize to other shapes or real 3D object data?**
>
> Response: Partially yes.
>
> The **contrastive question design** and the **WinoAcc-style metric** generalize naturally to other tasks and real 3D scenes. Any rule-based reasoning task with well-defined true/false outcomes can, in principle, adopt this framework to obtain robust chance-level baselines and confidence intervals.
>
> However, the **cognitively stratified question design** is more challenging to generalize. This is one reason we focus on cube net folding as the pivot task.
>
> For the cognitive stratification to work, the task should satisfy:
>
> **1. Ambiguous but well-structured symbolic content.**
> The 3D visual content must be rich enough to support multiple levels of questioning (from basic perception to high-level reasoning) while retaining a well-defined ground truth.
>
> **2. Shared images across cognitive levels.**
> Questions of different cognitive difficulty must be constructed over exactly the same images, so that perceptual biases are held constant and differences in performance can be attributed to reasoning rather than perception.
>
> **3. Multiple valid reasoning strategies constrained by 3D geometry.**
> To probe “imagination-level” evaluation, the task should admit multiple partial solutions that are all geometrically valid under Euclidean constraints (e.g., in cube net folding, an “L‑shaped” subset of the net that directly matches three visible faces provides a legitimate partial strategy). This enables us to test whether models discover and flexibly apply such strategies, instead of memorizing a single rigid reasoning chain.
>
> Because cube net folding naturally satisfies these properties, it serves as an effective pivot task for probing Euclidean priors in current VLMs.

---

### Official Review · Reviewer_7MGa · 2025-11-01

**Soundness:** 3
**Presentation:** 2
**Contribution:** 1
**Rating:** 4
**Confidence:** 5

**Summary:**

The paper introduces EucliFold, a synthetic benchmark for evaluating spatial reasoning in Vision-Language Models (VLMs) through cube-net folding tasks. The authors claim to decompose spatial reasoning into three “cognitive” levels (Perception, Operation, Imagination) and propose a “Winograd-style” evaluation metric to mitigate response bias. Experiments show that current VLMs perform well at perception-level tasks but fail at higher reasoning levels.

**Strengths:**

1. The dataset is well-structured and technically sound in its generation pipeline.

2. The introduction of Winograd-Style Accuracy is the most elegant part of the paper. By comparing model behavior across minimal-pair samples, it effectively neutralizes response bias and probes genuine understanding rather than superficial correlation.

3. The paper shows commendable experimental rigor through its ablations on prompt style, negative-sample design, and chain-of-thought prompting, which effectively expose bias and reasoning shortcuts in current VLMs.

**Weaknesses:**

**1. Lack of conceptual novelty :**
The entire setup—cube folding, synthetic evaluation of 3D reasoning, multi-level breakdown of spatial ability—has been explored extensively in prior benchmarks such as [1][2][3][4][5]. The paper’s “cognitive stratification” is essentially a re-labeling of well-known task complexity tiers (Mental rotation→Operation; Mental simulation → Imagination) without offering new theoretical insight or analysis into why current models fail.

**2. Superficial empirical contribution :**
 The experimental findings (VLMs are good at perception, bad at reasoning) replicate established results rather than extend them. There is no mechanistic or representational analysis explaining how or why geometric priors fail to emerge. The work therefore lacks diagnostic depth.

**3. Limited scope of evaluation:**
 Restricting all tasks to cube folding confines the conclusions to a very narrow spatial domain. The benchmark tests geometric pattern recognition more than true 3D spatial generalization.

[1] *11Plus-Bench: Demystifying Multimodal LLM Spatial Reasoning with Cognitive-Inspired Analysis*
[2] *SpatialViz-Bench: An MLLM Benchmark for Spatial Visualization*
[3] *DOES SPATIAL COGNITION EMERGE IN FRONTIER MODELS?*
[4] *Mind the Gap: Benchmarking Spatial Reasoning in Vision-Language Models*
[5] *Defining and Evaluating Visual Language Models' Basic Spatial Abilities: A Perspective from Psychometrics*

**Questions:**

**1. Overstated framing :**
The paper heavily relies on cognitive-science terminology (e.g., “autonomous imagination”) without actual modeling or measurement of cognitive processes. This creates the impression of interdisciplinary depth without substantive connection to cognitive theory.

**2. Missing key quantitative details :**
 Several sections (e.g., final accuracies in Table/Figure 1) leave placeholders (“X%”, “Y%”), making it difficult to judge significance. Statistical testing and ablation rigor are below the expected standard for ICLR.

**3. No fundamental insight or methodological innovation:**
 Beyond dataset generation and metric adaptation, the paper does not propose any new algorithmic or representational approach to probing Euclidean priors. It therefore reads more as an engineering replication than a scientific advancement.

---

> ### Author Response · Authors · 2025-12-04
>
> We thank the reviewer for the insightful comments. We address each concern in detail below.
>
> **W1: Lack of conceptual novelty**
>
> Response: We appreciate the reviewer’s feedback on our use of cognitive terminology and agree that our initial presentation may have been misleading. We acknowledge that the general notion of “cognitive abilities” is not novel; we adopt these terms primarily for their simplicity, familiarity, and communicative convenience.
>
> Our motivation for a cognitive-stratified design stems from the hierarchical nature of the abilities actually required to solve difficult spatial reasoning problems, rather than from an attempt to propose a new cognitive taxonomy. We are not cognitive science experts and do not aim to exhaustively cover all possible (or all classical) cognitive abilities (such as those summarized in [3], as the reviewer notes). Instead, our goal is to leverage this hierarchy to isolate higher-level spatial abilities and thereby provide quantitative measurements for probing possible Euclidean priors in VLMs.
>
> Following the reviewer’s suggestion, we have revised the manuscript to more clearly explain how our task hierarchy relates to existing cognitive studies and to avoid overstating conceptual novelty on the cognitive side.
>
> **W2: Superficial empirical contribution**
>
> Response: We agree that the earlier version of the manuscript did not sufficiently highlight the distinct empirical contribution of EucliFold. In the revision, we clarify and emphasize that:
>
> EucliFold is, to our knowledge, the first spatial reasoning dataset to provide a strictly controlled chance-level evaluation with explicit confidence intervals, via a large number of balanced contrastive pairs. This allows robust detection of “near-zero” emergent ability and reduces the risk of spurious above- or below-chance results due to dataset bias.
> The hierarchical task design enables cross-modal and cross-level comparisons that reveal the partial-solution nature of current VLMs: they often succeed at lower-level or perception-dominated tasks but fail to generalize to higher-level, imagination-intensive tasks. This provides concrete evidence that current SOTA VLMs typically learn and deploy partial solutions rather than a fully generalizable 3D Euclidean prior.
> These points are now explicitly discussed and contrasted with prior work in the revised manuscript.
>
> **W3: Limited scope of evaluation**
>
> Response: We agree that EucliFold has a limited task scope, and this is a deliberate consequence of its diagnostic focus. To obtain trustworthy confidence intervals and reliable chance-level baselines, we use at least 128 contrastive question pairs for each problem type, resulting in over 8k questions per model. This makes each full evaluation relatively expensive, which constrains the number and variety of tasks we can include in a single benchmark.
>
> Moreover, as discussed in our response to Reviewer tg8N (Q2), not all possible spatial reasoning tasks satisfy the stringent requirements for this diagnostic framework (e.g., balanced contrastive structure, shared perceptual distribution across cognitive levels, and multiple valid partial solutions under geometric constraints). Cube net folding is one of the few tasks that naturally fits these criteria, which is why we focus on it as a pivot task for probing Euclidean priors.

---

### Official Review · Reviewer_U2rQ · 2025-11-01

**Soundness:** 3
**Presentation:** 3
**Contribution:** 3
**Rating:** 6
**Confidence:** 2

**Summary:**

The paper introduces EucliFold, a benchmark designed to probe 3D Euclidean spatial reasoning in Vision-Language Models (VLMs). It constructs synthetic cube-folding tasks and proposes a cognitively-stratified evaluation decomposing spatial reasoning into three hierarchical levels:
1. Perception – grounding visual inputs into spatial representations,
2. Operation – manipulating these representations per instructions,
3. Imagination – autonomous problem solving under spatial constraints.

Using Winograd-style minimal-pair contrastive evaluation, the benchmark aims to reduce bias and disentangle genuine geometric reasoning from statistical pattern matching. Experiments across major VLMs (GPT-4o, Gemini, Claude, Qwen, InternVL) show strong perception-level accuracy but near-chance performance on operation and imagination levels, suggesting current VLMs lack robust internal 3D spatial priors.

**Strengths:**

Unique Task Design: 3D Euclidean Folding under Controlled Symmetry
- The use of cube-net folding in Euclidean space is mathematically precise, cognitively interpretable, and generalizable.
- The dataset generation pipeline (24 rotational groups × 11 nets × mutation filtering) ensures exhaustive coverage of geometric configurations while minimizing bias — a notable methodological innovation.

Bias-Resistant Evaluation Methodology
- The adaptation of Winograd-style contrastive accuracy to visual–spatial reasoning is new and impactful.
- It isolates genuine geometric understanding from spurious response tendencies or dataset artifacts, addressing a major weakness in prior spatial-reasoning evaluations.

High Relevance and Clarity
- Addresses a timely and important question: Do VLMs possess emergent 3D Euclidean priors?
- Writing and figures are clear, well-structured, and easy to reproduce, strengthening its potential as a community benchmark.

**Weaknesses:**

Limited Model-Level Diagnosis
- The paper identifies deficits but doesn’t probe why models fail

Lack of Cross-Domain Validation
- It remains unclear whether EucliFold performance correlates with or transfers to other spatial benchmarks (LEGO-Puzzles, 3DSRBench)

**Questions:**

1. Have you tested whether models fine-tuned on EucliFold exhibit better generalization to real-world or naturalistic 3D reasoning tasks?

2. Did you analyze intermediate visual or multimodal representations (e.g., via attention maps or feature embeddings) to understand where VLMs fail — at perception, transformation, or integration stages?

3. Do you envision EucliFold as a diagnostic benchmark only, or also as a training dataset for spatially grounded VLMs?

---

> ### Author Response · Authors · 2025-12-04
>
> We thank the reviewer for the insightful comments. We address each point below.
>
> **W1: Limited Model-Level Diagnosis**
>
> Response: We have added more fine-grained, model-level “zoom-in” analyses (Appendix H) to better characterize where and how different VLMs succeed or fail on EucliFold.
>
> **W2: Lack of Cross-Domain Validation**
>
> Response: Due to the “emergent probing” nature of EucliFold, it is difficult to define a smooth, quantitative metric that correlates directly with performance on comprehensive 3D reasoning datasets such as LEGO-Puzzles or 3DSRBench. Instead, we compare patterns of strengths and failure modes across domains. This comparison suggests that EucliFold captures aspects of 3D Euclidean reasoning that are not fully reflected in those existing benchmarks.
>
> **Q1 & Q3: Have you tested whether models fine-tuned on EucliFold exhibit better generalization? Do you envision EucliFold as a diagnostic benchmark only?**
>
> Response: We currently position EucliFold primarily as a diagnostic benchmark. It is designed to probe whether models possess robust 3D Euclidean priors and to reveal the partial solutions they might rely on.
>
> While EucliFold could in principle be used for RL-based post-training (e.g., using GRPO) once VLMs achieve significantly above-chance performance, our main goal is not to train models to exploit a single fixed strategy. The cube net folding task is inherently a multi-solution problem: many different valid reasoning strategies can lead to correct answers. If we supervise or hard-code a specific intermediate reasoning chain, models may overfit to one program-like solution rather than developing flexible, human-like geometric intuitions.
>
> Our empirical findings indicate that current VLMs lack a generalized 3D Euclidean prior and instead rely on fragmented, task-specific partial solutions. This suggests that other existing datasets may also be insufficient to induce or reveal truly general 3D Euclidean reasoning. EucliFold is therefore intended as a focused probe of these underlying priors, rather than a dataset for teaching a single canonical reasoning procedure.
>
> **Q2: Did you analyze intermediate visual or multimodal representations to understand where VLMs fail?**
>
> Response: We did not conduct a systematic analysis of intermediate visual or multimodal representations. This is infeasible for most closed-source models, where internal activations are inaccessible. For open-source models, performing a full re-evaluation with representation logging would be computationally expensive. Instead, we provide several complementary behavioral metrics (e.g., error patterns across cube types, orientations, and cognitive levels) that indirectly indicate where models are likely to fail. These analyses are detailed in the revised manuscript.

---

### Official Review · Reviewer_r6Nd · 2025-11-01

**Soundness:** 3
**Presentation:** 2
**Contribution:** 2
**Rating:** 4
**Confidence:** 4

**Summary:**

This paper introduces EucliFold, a new visual QA benchmark designed to probe the 3D spatial reasoning capabilities of VLMs. EucliFold is built on synthetic cube net folding tasks.
The paper's primary contributions are:
 - A "cognitively-stratified" evaluation framework that decomposes spatial reasoning into three hierarchical levels: Perception, Operation, and Imagination
 - A new synthetic dataset for these tasks, generated with a controlled pipeline to create 2D nets and 3D cube views.
 - A "Winograd-style accuracy" metric that uses minimal-pair contrastive samples to mitigate VLM response biases and isolate genuine reasoning from statistical pattern matching.

Using this framework, the authors evaluate a suite of VLMs. The key finding is that while models show reasonable performance on Perception tasks, their abilities degrade significantly on Operation tasks and fail almost completely on Imagination tasks. This suggests that current VLMs rely on superficial 2D pattern matching rather than possessing robust, generalizable 3D geometric priors.

**Strengths:**

1. Cognitive Framework: The cognitively-stratified decomposition into Perception, Operation, and Imagination is a major strength. This framework is well-motivated by cognitive science  and allows for a much more nuanced analysis than a single "spatial reasoning" score. It helps pinpoint where and why models are failing, moving from "can they see?" to "can they manipulate?" to "can they solve?"

2. Evaluation: The authors show a deep awareness of the common pitfalls in VLM evaluation. The use of minimal-pair contrastive samples and the "Winograd-Style Accuracy" metric  is an excellent choice. This bias-resistant approach helps ensure that the results reflect genuine reasoning gaps rather than just dataset artifacts or model response biases , which is a common weakness in other benchmarks. The ablation studies (e.g., on "easy negative samples" ) further validate their methodology.

**Weaknesses:**

1. Insufficient Comparison to Related Work: The paper’s novelty is limited as it fails to adequately differentiate itself from or compare against several existing spatial reasoning benchmarks (e.g., SPACE, OmniSpatial, SpatialViz-Bench, STARE). A direct comparison is needed to justify this benchmark's unique contribution.

2. Missing Evaluation of Latest SOTA Models: The evaluation is incomplete. To make a strong claim about the limitations of current VLMs, the paper must include results from the latest SOTA models, such as GPT-5, Gemini 2.5 Pro, the Claude 4 series, and InternVL 3.5.

3. Lack of Detailed Results Tables: The paper relies almost exclusively on figures to present results. This makes it difficult to ascertain precise numerical scores. The paper would be much stronger if it included dedicated results tables, especially for the Operation and Imagination tasks, rather than only referring to Figure 1.

4. Superficial Error Analysis: The analysis lacks depth. While the paper identifies that models fail at Operation and Imagination, it does not investigate why. A deeper qualitative analysis of the failure modes is missing.

5. Mismatch between "Euclidean" Claims and Task: The "EucliFold" title is a misnomer. The task is limited to cubes, which are highly constrained, discrete objects, and may not adequately probe an understanding of general 3D Euclidean space. Furthermore, the main paper omits details on how prompts were transformed into "Euclidean terms" for the "Euc" ablation study, making this claim hard to verify.

**Questions:**

1. Following on from Weakness 2, what is the performance of the latest SOTA models (e.g., Gemini 2.5 Pro, Claude 4-series, Qwen3-VL) on your benchmark?

2. The data generation uses 50 distinct cubes, each with its 24 possible 3D representations. Since the dataset is synthetic, what is the reasoning for this design (low cube variety, high rotational variety) versus using a much larger set of unique cube patterns?

3. Paper mentions "near-perfect human performance" and details the human study in Appendix E. What were the precise, aggregated human accuracy scores for each of the three cognitive levels?

4. The scaling plots (e.g., Figure 4) mix different model families (InternVL, Qwen) on the same x-axis ("Model Size"). This is not a valid scaling analysis, as it confounds scale with architecture. Can you provide scaling plots using a single, consistent model family?

---

> ### Author Response · Authors · 2025-12-04
>
> We thank the reviewer for the insightful comments. We address each concern in detail below.
>
> **W1: Insufficient Comparison to Related Work**
>
> Response: We have revised the manuscript to include a more detailed comparison with related work. The corresponding changes are highlighted in blue. We emphasize that the most significant difference between our work and prior studies lies in the motivation and problem formulation.
>
> **W2 & Q1: Missing Evaluation of Latest SOTA Models**
>
> Response: We have additionally evaluated the most recent state-of-the-art VLMs (GPT-5, Claude-4-Sonnet, and Gemini-2.5-Pro). As shown in the updated table, their performance is not drastically higher than that of their immediate predecessors.
>
> **W3 & Q3: Lack of Detailed Results Tables**
>
> Response: We have added detailed tables to the manuscript reporting numerical results, including both model performance (Appendix F) and human performance (Appendix G).
>
> **W4: Superficial Error Analysis**
>
> Response: We have expanded the error analysis (Appendix H). And adjust the Introduction part to highlight three distinct advantages of our framework, as well as the novel empirical findings it enables.
>
> **W5: Mismatch Between “Euclidean” Claims and Task**
>
> Response: We appreciate the reviewer for pointing this out. We agree that the cube-folding task and the EucliFold dataset do not provide a fully comprehensive evaluation of models’ Euclidean priors. However, probing Euclidean priors is intrinsically challenging, and to our knowledge no existing dataset can quantitatively evaluate them in a well-rounded manner. EucliFold is a pioneering attempt toward probing potential Euclidean priors, using cube folding as a classical and well-defined test case. We therefore combine these two aspects and name the dataset EucliFold. We intentionally avoid a narrower name such as “CubeFold,” because our goal is not merely to benchmark general cube net folding performance, but to explore Euclidean reasoning more broadly through this task.
>
> **Q2: The data generation uses 50 distinct cubes, each with its 24 possible 3D representations. Since the dataset is synthetic, what is the reasoning for this design?**
>
> Response: During the development of EucliFold, we anticipated that some SOTA models might learn partial solutions specific to certain cube nets (e.g., the well-known T-shaped net) or certain combinations of color patterns (e.g., spurious correlations between color and arrow direction). To systematically probe such behaviors, for each distinct cube we enumerate all 24 possible 3D orientations and generate questions across multiple cognitive levels, resulting in
>  questions per cube.
>
> All of these questions share the same randomly selected image set, and thus the same perceptual and pattern-related biases. This design is crucial for identifying whether models exploit partial solutions or spurious correlations. Unfortunately, most current SOTA models cannot even reliably leverage such partial solutions, which prevents us from conducting the fine-grained analysis we originally envisioned and may have caused confusion about the motivation behind this design.
>
> **Q4: The scaling plots (e.g., Figure 4) mix different model families.**
>
> Response: We apologize for the confusion and have clarified the table and figure notes. The scaling plots only include six models from the InternVL-2.5 family. We focus on this family because it is open-source, spans a wide range of scales, and is architecturally consistent. As shown in the updated table, this allows a cleaner analysis of scaling behavior within a single model family.

---

### Meta-Review · Area_Chair_625z · 2025-12-24

**Summary:**

This paper proposes a new framework for spatial reasoning, focusing on perception, operation, and imagination, with a new benchmark for evaluation. Multiple reviewers commented on some limited novelty as well as evaluation and diagnosis. Given that there exists a rich set of spatial reasoning benchmarks that are individually structurally motivated, there is a concern on what is the exact novelty and necessity of this framework that warrants inclusion. Furthermore, such a benchmark should have a rigorous analysis of where current models fail, why they fail, as well as some direction for next steps towards improving (is there a fundamental challenge or can it be mitigated with some intuitive techniques). As such, I cannot recommend acceptance.

**Reviewer Concerns:**

- Marginal novelty and comparison to related work: There exist a number of different spatial reasoning frameworks that are all structurally motivated.
- Limited evaluation with models: The rebuttal has introduced new experiments with more models.
- Limited analysis of errors and diagnosis: Given the large number of spatial benchmarks, it should be necessary to characterize in depth how and why the current weaknesses exist; and ideally, suggest some approach to mitigation. The rebuttal includes some limited error analysis, but it is still insufficient.

**Reviewer Scores:**

I do not believe that any of the reviewers would change their scores. Multiple reviewers commented on common concerns: 1) novelty; 2) limited spread of models; 3) limited deep evaluation. While the 2nd concern has been reasonably addressed in rebuttal, the first and third concern have not.

---

### Decision · Program_Chairs · 2026-01-26

Reject